# Suit the Remedy to the Case—The Effectiveness of COVID-19 Nonpharmaceutical Prevention and Control Policies Based on Individual Going-Out Behavior

**DOI:** 10.3390/ijerph192316222

**Published:** 2022-12-04

**Authors:** Siqing Shan, Feng Zhao, Menghan Sun, Yinong Li, Yangzi Yang

**Affiliations:** 1School of Economics and Management, Beihang University, Beijing 100191, China; 2Beijing Key Laboratory of Emergency Support Simulation Technologies for City Operation, Beijing 100191, China

**Keywords:** nonpharmaceutical epidemic prevention and control policies, policy intensity, policy evaluation, going-out needs, simulation method, SEIR

## Abstract

Nonpharmaceutical policies for epidemic prevention and control have been extensively used since the outbreak of COVID-19. Policies ultimately work by limiting individual behavior. The aim of this paper is to evaluate the effectiveness of policies by combining macro nonpharmaceutical policies with micro-individual going-out behavior. For different going out scenarios triggered by individual physiological safety needs, friendship needs, and family needs, this paper categorizes policies with significant differences in intensity, parameterizes the key contents of the policies, and simulates and analyzes the effectiveness of the policies in different going-out scenarios with simulation methods. The empirical results show that enhancing policy intensity can effectively improve policy effectiveness. Among different types of policies, restricting the times of going out is more effective. Further, the effect of controlling going out based on physiological safety needs is better than other needs. We also evaluate the policy effectiveness of 26 global countries or regions. The results show that the policy effectiveness varies among 26 countries or regions. The quantifiable reference provided by this study facilitates decision makers to establish policy and practices for epidemic prevention and control.

## 1. Introduction

Since the outbreak of coronavirus disease 2019 (COVID-19), the epidemic has spread rapidly and brought huge losses and threats to the lives of people globally [1]. COVID-19 has received extensive attention from the public and academic domains. Many researchers have carried out a substantial body of studies on COVID-19, which mainly focus on (1) research on the risk factors for virus infection [2,3,4,5,6]; (2) prediction models for epidemic transmission [7,8,9,10,11,12]; (3) the effects of COVID-19 on the economy [13,14,15]; (4) the effects of COVID⁃19 on people’s psychological health [16,17,18,19,20,21]; and (5) nonpharmaceutical epidemic prevention and control policies.

To control the epidemic, governments of various countries have successively implemented many nonpharmaceutical prevention and control policies [22]. According to statistics, more than 190 countries or regions have released over 13,000 policies [23], including (1) policies on case-driven measures such as testing, tracing, and distancing; (2) policies on personal prevention such as reducing face touching, wearing masks in crowded or enclosed spaces, and physical distancing; and (3) policies on social distance such as home order, lockdown, and bans [24]. Furthermore, studies have added evidence that policies such as lockdown [25,26], physical distancing [27,28,29], and wearing masks in crowded or enclosed spaces [30,31,32] play important roles in slowing down the spread of the epidemic. From the current practices of epidemic prevention and control, many countries and regions have implemented policies with different contents and intensities based on their own public health and socioeconomic status [23].

Evaluating the effectiveness of nonpharmaceutical policies is essential for policy design and implementation. Many scholars have studied the effectiveness of policies. Based on a nonpharmaceutical intervention module and Metropolis–Hastings sampling, Zhao et al. proposed a susceptible-infected-recovered-vaccinated (SIRV) model, the results showed that stringent nonpharmaceutical interventions (NPIs) are the key to controlling the COVID-19 epidemic [33]. With a discontinuous difference-in-difference approach, Deng et al. found that adopting rigid NPIs could reduce the number of new COVID-19 cases by 10.8% in China, and that contact tracing is much more effective than public information provision and social distancing [34]. Wieland assessed the effectiveness of German interventions such as “lockdown” and found that the effectiveness of most interventions remained questionable [35]. Naimark et al. developed an agent-based transmission model to estimate the impact on the number of COVID-19 cases of schools being open or closed and community-based NPIs. The findings demonstrated that community-based NPIs were more effective [36]. Lee et al. proposed an agent-based simulation model to assess the effectiveness of NPIs, including social distancing, face mask use, school closure, testing, and contact tracing. The results showed that infections could remain low and other NPIs could be relaxed when face mask use was at least 75% [37]. With the simulation method, Ge et al. studied the effectiveness of social distancing, contact tracing, and case isolation. They suggested that contact tracing merits further attention to achieve population-level control of a second-wave epidemic [38]. Cho found that strict lockdown measures were important in limiting the spread of the COVID-19 infection using a synthetic control approach [39]. Ilhan examined NPIs applied in Turkey and found that restrictions implemented in Turkey such as isolation, quarantine, and contact tracing could flatten of the epidemic curve [40].

The above studies explored the impact of policy intensity and types on epidemic prevention and control based on simulation models and mathematical models; however, there is still room for improvement. First, when considering combining different policies, how to formulate the intensity of each policy is worth further consideration. Second, the spread of the epidemic occurs in the process of individual going out, so the policy should also act on different links of the process. Classifying the policy based on individual going-out behavior and evaluating its effectiveness is more conducive to the formulation and implementation of the policy. Third, existing studies rarely distinguish the difference in social networks between individuals. The spread of COVID-19 in different social networks is different and worthy of further exploration. 

It is generally believed that if more policies are implemented and the intensity of each policy is stricter, epidemic prevention and control will be more effective. However, at the same time, such measures will have a greater effect on people’s daily lives. How to choose appropriate policies and reasonable policy intensities according to the actual situation requires a solution to the problem of policy comparison and evaluation. Specifically, how to quantitatively compare the effectiveness of different types of policies and how to accurately evaluate the effectiveness of certain policies under different intensities are the critical problems that currently challenge the formulation and implementation of epidemic prevention and control policies.

The aim of this paper is to solve the scientific problems mentioned above. From the perspective of individual going-out behavior, we built a model of individual going-out and epidemic spread using a simulation-based method. The spread of the epidemic and the infection process are simulated in various scenarios, and the infection curves under different scenarios are estimated. Therefore, policy effectiveness can be evaluated and compared. Except for the robustness test, all scenarios in the simulation process of this paper are composed of 2000 individuals.

## 2. Research Methods

### 2.1. Study Design

The mechanism diagram of individual going-out behavior is shown in Figure 1. We assume that individuals have three kinds of needs: family needs, friendship needs, and physiological safety needs. When individuals go out, they will contact a certain number of people, and in this process, there is a risk of the epidemic spreading further.

From the perspective of individuals going out, we have divided various nonpharmaceutical prevention and control policies into three types. The three types of policies play certain restrictive roles in different links. The first type is the policy of restricting going out (RGOUT), which affects individuals’ decisions on whether to go out or not, with the purpose of reducing unnecessary times of going out, such as a stay-at-home order and lockdown. The second type is the policy of restricting contact (RCONT), which acts on individuals after they go out and aims to reduce large-scale gatherings and minimize unnecessary contact, such as restrictions of mass gatherings. The third type is the policy of restricting infection (RINFE) when individuals go to crowded places. The purpose is to reduce the infection rate as much as possible through measures such as maintaining physical distance, reducing face touching, and wearing masks in crowded or enclosed spaces. The three types of policies have different effects on different links. To effectively prevent and control the epidemic in combination with the actual situation, it is necessary to select suitable policies and the rational intensity of policy implementation.

We propose a quantitative policy simulation comparison and evaluation model for COVID-19 epidemic prevention and control, combining the SEIR model, going-out behavior modeling, social networks, and other methods.

First, we use the classic disease transmission model (SEIR model) [41,42] and fully consider the characteristics of COVID-19. All individuals can be in one of the four states: susceptible, exposed (in the incubation period, showing no obvious symptoms but infectious), infectious (after the incubation period, showing obvious symptoms, and more infectious), and recovered (assumed to be no longer infected). When an individual with exposed or infectious status contacts a susceptible individual, there is a certain probability that the susceptible individual will be transformed into the exposed status. In addition, exposed individuals will change to the infectious state with a corresponding probability over time during the incubation period (assuming no asymptomatic infections). Infectious individuals will change to the recovered state with a corresponding probability over time.

Second, individuals’ subjective needs determine their behaviors [43], and the model proposed in this paper adequately considers their needs. According to Maslow’s hierarchy of needs, there are five different levels of needs: physiological needs, safety needs, social needs, esteem needs, and self-actualization needs [44]. During COVID-19, people still need to go to restaurants [45] and hospitals [46] for their basic needs. In addition, people still need to fulfill higher-order psychological needs [47]. Therefore, we selected physiological safety needs and social needs. Physiological safety needs mainly refer to the acquisition of things that are vital to basic survival and safety, such as water and food. Social needs are the needs for affection and belonging, which can be divided into friendship needs and family needs. Friendship needs refer to the need to integrate into the social circle, communicate with friends, and feel love and acceptance from friends. Family needs refers to the need to avoid problems such as loneliness and anxiety, involvement with family members and feeling care and belonging from families. The three kinds of need-driven behaviors are different, and the individuals they contact are also different. This means that the social networks generated by the three needs are different, which may lead to differences in the spread of the epidemic. The randomness of individual contact based on the three kinds of needs is different. Contact based on family needs is relatively fixed, and contact based on physiological safety needs is random. The randomness of contact based on friendship needs is somewhere in between.

Finally, we build the network based on three needs and simulate the relationship between family, friends, and strangers in the real world by networks. Based on the similarity of features such as geographic location, age, and blood relationship, people with higher family similarity are connected as family members; based on the similarity of features such as interest, occupation, income, mutual friends, and interaction, people with higher social similarity are connected as friends; for the construction of the stranger network, we assume that every individual has ten basic physiological safety needs. For each need, individuals with a high similarity are connected based on similarities of occupation, geographic location, and other features as the list of strangers under this need. The networks generated based on different needs have different characteristics: if individuals are identified by geographic location, the network formed by family needs has more clusters in form, which is similar to multiple families in real life, and the friend network is relatively scattered. A stranger network is more scattered, and there will be many hub nodes in the network (that is, nodes with a large number of connections in the network), similar to departments with key functions in real life, such as supermarkets and hospitals. These characteristics are also consistent with the randomness of individual contacts.

### 2.2. Social Network Construction

This paper builds a family network, a friend network, and a stranger network based on three needs of going out. First, the family network is built based on family needs. Each individual is assigned twenty features, and each feature is represented by an integer from 1 to 10. Then, the similarity between every two individuals is calculated by the Pearson correlation coefficient. The calculation method of the Pearson correlation coefficient is as follows:$$\;\rho_{X,Y}=\frac{cov\left(X,Y\right)}{\sigma_{X}\sigma_{Y}}=\frac{E\left[\left(X-\mu_{X}\right)\left(Y-\mu_{Y}\right)\right]}{\sigma_{X}\sigma_{Y}}$$

For each individual, the other individuals are sorted in descending order of similarity with the individual, and the top 20 individuals are taken as the individual’s family members. For each individual, the family list is fixed throughout the simulation process.

Next, the friend network is built based on friendship needs. Each individual is also assigned twenty features. Two of the features are the mutual friend ratio and the interaction ratio. To be more realistic, we construct a small-world network composed of 2000 individuals and suppose that each individual has 2 neighbors on average, and the random reconnection rate is 0.3. Subsequently, a weight in the interval of 0~100 is randomly generated for each connection in the network, which represents the interaction frequency of two individuals connected to each other. Finally, the mutual friend ratio and interaction ratio of each pair of individuals are calculated based on the network connection. The mutual friend ratio is the ratio of the number of mutual friends of the two to the number of all friends of the individual. The interaction ratio is the ratio of the interaction frequency of the two to the total number of interactions of the individual. The values of the mutual friend ratio and interaction ratio are in the range of 0~1. The remaining 18 features are represented by integers from 1 to 10. First, the weighted sum of the mutual friend ratio and the interaction ratio are calculated to obtain the social interest degree, and the weights of both are 0.5. Second, based on the remaining 18 features, the Pearson correlation coefficient is used to calculate the similarity between every two individuals as the interest similarity. Finally, the social interest degree and the interest similarity degree are weighted and summed to obtain the similarity between every two individuals, and the weight of both is 0.5. For each individual, the other individuals are sorted according to the similarity with the individual from high to low. After removing the members of the individual’s family list, the top 20 individuals are taken as the individual’s friend list. For each individual, the friend list is fixed throughout the simulation process.

Finally, a stranger network is built based on physiological safety needs. The twenty features of each individual are represented by an integer from 1 to 10. It is assumed that each individual has ten types of physiological safety needs, and each type of need is a 20-dimensional vector. For each need, the similarity between the other individuals’ feature vector and this need vector is calculated by the vector cosine, and the similarity is sorted from high to low. After removing the members of the individual’s family and friends list, the top 20 individuals are taken as the individual’s list of strangers under this need. Each individual has 10 stranger lists. Each time an individual goes out, one need is randomly selected and corresponds to one stranger list. For each individual, in the entire simulation process, the stranger list changes with the different physiological safety needs of going out.

### 2.3. Simulation Model Establishment

Based on the classic SEIR model, considering the actual situation of COVID-19, our model assumes that individuals may be in one of the four different states: susceptible (S), exposed (E), infectious (I), or recovered (R). The virus can spread through contacts, and the specific process is illustrated in Figure 2.

The simulation process is as follows:
A certain number of individuals are randomly selected as the initial infected individuals ($n_{initial}$), and their state (E or I) and the time in that state ($T_{exposed}$ or $\;T_{infectious}$) are initialized. The remaining individuals are initialized to be susceptible. This paper does not consider asymptomatic infections.Every day, the times of going out for all individuals are generated according to the maximum times of going out per day (Daily Go Out Times Threshold), which is determined by the RGOUT policy. During each time of going out, the number of contacts is generated according to the maximum number of contacts at a time (Daily Everytime Contact Threshold), which is determined by the RCONT policy. At each contact, the infection rate (Infection Rate) is determined by the RINFE policy and individual states.Every day, an individual i is randomly selected from the list of individuals who can go out on that particular day, with the remaining times of going out not zero. Then, a type of needs is randomly selected. The individual list of contacts is selected from the corresponding network with the probability $p_{contact}$. Individual *i* makes contact with each individual in the list.The infected individual (E or I) infects a susceptible individual (S) at a certain rate ($p_{infection}$). After completing a time of going out, step 3 is repeated until all of the day’s going out times of all individuals are used up.An exposed individual (E) will transform into the infectious state (I) with probability $p_{E-I}$, which is related to the incubation period. After being infectious for a certain period, the infectious individual recovers with probability $p_{I-R}$. The probability is influenced by the infection period. Once recovered, individuals will no longer be infected.Steps 2, 3, 4, and 5 are repeated. The simulation is terminated until the state of all individuals is in S or R.

Note that individuals in both the exposed and infectious states are infectious. Generally, the infection rate of the exposed individual is lower than that in the infectious state. This paper assumes that the infection rate of the exposed individual is half of that in the infectious state [48].

When selecting individuals to be contacted for each time of going out, the selection probability is defined as $p_{contact}$. In the network of family and friends, $p_{contact}$ is the similarity between individuals, as calculated by the Pearson correlation coefficient based on the features of the individuals. The more similar the two individuals are, the higher the probability of contact is. In the stranger network, $p_{contact}$ represents the matching degree of the physiological safety needs between two individuals. It is calculated by cosine similarity based on the need vector of the individual and the feature vector of other individuals. Here, it is assumed that the higher the matching degree between two individual features, the higher the probability of contact is.

The probability of an individual changing from an exposed state to an infectious state ($p_{E-I}$) is calculated by the following formula:$$p_{E-I}=\frac{T_{exposed}}{T_{MaxExposed}}$$
where $T_{MaxExposed}$ represents the longest exposure time; that is, when the longest exposure time is reached, the exposed individual will automatically become the infectious state.

The probability of an individual changing from an infectious state to a recovered state ($p_{I-R}$) is calculated as:$$p_{I-R}=\frac{T_{infectious}-T_{threshold}}{T_{MaxInfectious}-T_{threshold}}$$
where $T_{threshold}$ represents a threshold after which the individual has the probability of transferring from infectious to recovered. $T_{MaxInfectious}$ represents the longest infectious time. When the longest infectious time is reached, the infectious individual will automatically change to the state of recovery and no longer be infected.

### 2.4. Quantitative Design Strategies for Prevention and Control Policies

In this paper, the policies for epidemic prevention and control are divided into three types: RGOUT policy, RCONT policy, and RINFE policy. RGOUT policies such as stay-at-home orders and lockdowns mainly limit the times of going out. In the simulation model, the policy is quantified as a parameter named Daily Go Out Times Threshold. Different values of this parameter correspond to different intensities of the RGOUT policy. Many countries and regions (Israel, Germany, China, etc.) instructed not to go out unless absolutely necessary and some have restricted the specific number of times of going out: Colombia (Medellin) limited one member of each family to be able to go out twice a week (on the 2 April 2020); and Uzbekistan, Namangan and Uchqo’rg’on Districts of Namangan Region instruct residents to go out once a day (on the 13 April 2020) [23]. RCONT policies, such as restrictions on mass gatherings, mainly limit the number of contacts while people go out. In the simulation model, the policy is quantified as a parameter named Daily Every Time Contact Threshold. Different values of this parameter correspond to different intensities of the RCONT policy. Many countries and regions have imposed restrictions on the number of people gathered. For example, the Australian Government announced gatherings will be restricted to two persons only (on the 29 March 2020); Georgia banned gatherings of more than three people (on the 31 March 2020); and in Indonesia, the Jakarta provincial government issued a restriction on mass gatherings of up to 5 people (on 10 April 2020) [23]. RINFE policies, such as reducing face touching and wearing masks in crowded or enclosed spaces, mainly standardize people’s personal health protection. In the simulation model, the policy is quantified as a parameter called Infection Rate, which refers the per-contact transmission probability that an infectious individual transmits the virus to a susceptible individual. Different values of this parameter correspond to different intensities of the RINFE policy. The infection rate is between 0.002–0.1 [49].

The quantitative design strategies for the three types of prevention and control policies are indicated in Table 1.

### 2.5. Simulation Parameter Description

Except for the robustness test, the number of individuals in the simulation process in this paper is set as 2000. In the robustness test, the numbers of individuals are 4000 and 5000, respectively. In addition, the other parameters are as follows: the initial number of infected individuals is 3, the longest exposed time is 14 (the incubation period of ranges typically from 2–14 days [50]), the threshold for starting the transformation from the infected state to the recovered state is 10, and the longest infected time is 21 (infectious period is estimated between 10–20 days [50]).

### 2.6. Experimental Scenario Design Strategy

The experimental scenario design strategy mainly consists of two aspects. One is to set different policy intensities by adjusting the parameter values controlled by three types of policies, and the other is to set different kinds of going-out needs. In this way, multiple scenarios can be combined based on different going-out needs, different policies, and different policy intensities.

In this paper, a variety of simulation experiments are conducted. The first set of experiments consists of basic experiments based on three kinds of going-out needs (Table 2, experiments 1-1, 1-2, and 1-3). By setting the intensity of the three policies to be loose, moderate and strict, their effectiveness in epidemic prevention and control could be compared.

Then, a second set of variation experiments based on three kinds of going-out needs (Table 3, Experiments 2-1 to 2-6) is set up to compare the differences in policy effectiveness by fixing the intensity of two policies in a loose state and varying the intensity of the other policy. Similarly, a third set of variation experiments based on three going-out needs (Table 3, Experiments 3-1 to 3-6) is set up to compare the differences in policy effectiveness by fixing the intensity of two policies under strict conditions and varying the intensity of the other policy.

Finally, a fourth set of basic experiments based on different kinds of going-out needs (Table 4, Experiments 4-1 to 4-3) is set up to explore the impact of different needs on policy effectiveness.

The experiments in all scenarios are simulated 50 times, and the average value is taken.

### 2.7. Comparison and Evaluation Methods of Actual Epidemic Prevention and Control Policies

To cross-compare the real new daily infection data from different countries and regions around the world with the new daily infection data obtained by simulation, an improved curve similarity indicator based on dynamic time warping (DTW) is proposed in this paper to allow the actual effectiveness of epidemic prevention and control policies could be analyzed and evaluated.

Dynamic time warping (DTW), a method of time warping using dynamic programming (DP), can automatically scale time series of different lengths and complete the similarity calculation of time series with different lengths and rhythms. It was proposed by Japanese scholar Itakura in the 1960s.

Suppose there are two time series, $Q\;=\left\{q_{1},q_{2},\cdots ,q_{n}\right\}$ and $C\;=\left\{c_{1},c_{2},\cdots ,c_{n}\right\}$, with lengths n and m, respectively. The similarity calculation process between the two time series Q and C using dynamic time warping is as follows:
An n ∗ m matrix $D\;=\;\{d_{ij}|d_{ij}=\;\mathrm{dist}\left(q_{i},c_{j}\right),i\;=1,2,\cdots ,n;\;j\;=1,2,\cdots ,m\}$ is constructed where dist(x,y) represents the distance calculation function and the Euclidean distance based on a one-dimensional vector is used, namely, $\mathrm{dist}\left(q_{i},c_{j}\right)=\sqrt{{\left(q_{i}-c_{j}\right)}^{2}}=\left|q_{i}-c_{j}\right|$;Using a dynamic programming algorithm, the shortest path from $d_{11}$
to $d_{nm}$ in matrix D is searched. At position $d_{ij}$, there are only three cases for the path search direction, that is, $d_{\left(i+1\right)j}$, $\;d_{i\left(j+1\right)},d_{\left(i+1\right)\left(j+1\right)}$;According to dynamic programming, a shortest path from $d_{11}$ to $d_{nm}$ that satisfies the path search direction is obtained, and it is named the warping path, expressed by W. The k-th element of W is defined as $W_{k}={\left(q_{i},c_{j}\right)}_{k}$, $W\;=w_{1},w_{2},\ldots ,\;w_{k},\ldots ,w_{K}$, where max(m,n) ≤ K < m + n −1. The calculated shortest path length is output as the similarity; that is, $\mathrm{DTW}\left(Q,C\right)=\sum_{k=1}^{K}dist\left(w_{k}\right)$, which is the dynamic time warping similarity between the time series Q and C.

Considering that the similarity obtained by the traditional dynamic time warping method is accumulated by the point-to-point distance of two time series, its output value is affected by both the number range and the length of the time series itself. Affected by the above two factors, the similarity obtained by the traditional dynamic time warping method is unsuitable for achieving horizontal comparisons between fixed sequences and other sequences with different lengths and ranges. In response to this problem, based on dynamic time warping, we propose an improved curve similarity calculation method that is suitable for this scenario. More specifically, the calculation consists of the following steps:

Suppose there are two time series $Q\;=\left\{q_{1},q_{2},\cdots ,q_{n}\right\}$ and $C\;=\left\{c_{1},c_{2},\cdots ,c_{n}\right\}$, with lengths of n and m, respectively,

The sequence of Q and C is normalized into the 0–1000 interval with min-max normalization, that is,
$Q_{normed}=\{\left(q_{1}^{\prime },q_{2}^{\prime },\cdots ,q_{n}^{\prime }\right)|q_{i}^{\prime }=\frac{\left(q_{i}-q_{min}\right)*1000}{\left(q_{max}-q_{min}\right)},q_{max}=max^{i}\left(q_{i}\right),q_{min}=min^{i}\left(q_{i}\right),i\in 1,2,\cdots ,n\}$, and $C_{normed}$ can be obtained in the same way.The DTW similarity of $Q_{normed}$ and $C_{normed}$ is calculated with dynamic time warping.The maximum value of the time series length is used to normalize $\mathrm{DTW}\left(Q_{normed},C_{normed}\right)$, that is, sim(Q,C)=DTW(Qnormed,Cnormed)max(m,n). Then, the final sequence similarity between Q and C is obtained, denoted as sim(Q,C).

## 3. Results

This paper set up various experimental scenarios by simulation, including basic experiments based on three kinds of going-out needs, variation experiments based on three kinds of going-out needs, and basic experiments based on different kinds of going-out needs. We obtained the number of infections over time during the spread of the epidemic. Then, the infection curves under different experimental scenarios could be estimated to compare and evaluate the prevention and control effectiveness of different epidemic prevention and control policies. Here, the policy effectiveness refers to the effect of the policy in slowing down the spread of the epidemic, which can be reflected in the flatness of the infection curve. The flatness of the infection curve is mainly related to the peak value of infections and the epidemic duration. Therefore, to quantitatively compare the flatness of different infection curves, the flatness index of the infection curve is defined to serve as a basis for comparing the effectiveness of different policies. The definition of the flatness index of the infection curve is as follows:flatness index of infection curve = epidemic duration/peak value

A larger flatness index of the infection curve indicates a smoother corresponding infection curve and the slower spread of the epidemic. This also means that the effectiveness of the prevention and control policy is better.

The peak value of infections, peak arrival time, epidemic duration, and flatness index of the infection curve corresponding to all experimental scenarios are shown in Appendix A.

### 3.1. Results of Basic Experiments Based on Three Kinds of Going-Out Needs

There are significant differences in the effectiveness of all three types of policies. When all three policies are in a strict state, the effect of epidemic prevention and control is the best. The flatness indices of the infection curve are 37.93 times and 145.10 times those of the three policies in the moderate and loose states, respectively.

Figure 3 shows the infection curves of the first set of experiments (experiment 1-1 to experiment 1-3). The *x*-axis represents time (days), and the *y*-axis indicates the number of infected individuals (E or I). The blue curve represents the experimental scenario when the intensity of the three policies is strict (that is, the parameters controlled by the three policies are at the minimum). The curve is the flattest, with a flatness index of 3.5840, a peak value of 119.42, and an epidemic duration of 428 days. The green curve represents the experimental scenario when the intensity of the three policies is moderate (that is, the parameters controlled by the three policies are intermediate values). The curve is slightly flat, with a flatness index of 0.0945, a peak value of 1259.68, and an epidemic duration of 119 days. The red curve represents the experimental scenario when the intensity of the three policies is loose (that is, the parameters controlled by the three policies are maximized). The curve is very steep, with a flatness index of 0.0247, a peak value of 1944.56, and an epidemic duration of 48 days. Compared with the infection curves when the three policies are at moderate and loose intensity, the infection curve when the three policies are in a strict state can be flattened: the peak values drop by 90.52 and 93.86%, the peak arrival times are delayed by 237.78 and 623.81%, and the epidemic durations are extended by 259.66 and 791.67%, respectively.

Overall, the strict control of policy has a more obvious inhibitory effect on decreasing the peak value, delaying the peak arrival time and extending the epidemic duration than the loose and moderate control.

### 3.2. Results of Variation Experiments Based on Three Kinds of Going-Out Needs

The effect of the RGOUT policy is relatively better in the three types of epidemic prevention and control policies. When the other two types of policies are loose, increasing the intensities of the RGOUT, RCONT, and RINFE policies from loose to moderate and strict states, the flatness index of the corresponding infection curve could be increased by 85.27, 208.45, 38.69, 135.43, 15.61, and 56.36%, respectively. When the other two types of policies are strict, increasing the intensities of the RGOUT, RCONT, and RINFE policies from a loose to a moderate state, the flatness index of the corresponding infection curve could be increased by 357.73, 103.49, and 65.19%, respectively.

In the second set of experiments (experiment 2-1 to experiment 2-6), policy effectiveness is compared by fixing the intensity of two types of policies in the loose state and varying the intensity of the other policy. When the intensities of the RCONT and RINFE policies are loose and the intensity of the RGOUT policy is increased from loose to moderate and strict, the infection curve is gradually flattened (Figure 4): the peak values decrease by 13.41 and 29.08%, the peak arrival times are delayed by 47.62 and 90.48%, and the epidemic durations are prolonged by 60.42 and 118.75%, respectively. Similarly, when the intensities of the RGOUT and RINFE policies are loose and the intensity of the RCONT policy is increased from loose to moderate and strict, the infection curve is gradually flattened (Appendix A), the peak values decrease by 5.36 and 19.47%, the peak arrival times are delayed by 19.05 and 61.90%, and the epidemic durations are prolonged by 31.25 and 89.58%, respectively. When the intensities of the RGOUT and RCONT policies are loose and the intensity of the RINFE policy is increased from loose to moderate and strict, the infection curve is gradually flattened (Appendix A), the peak values decrease by 2.69 and 9.40%, the peak arrival times are delayed by 9.52 and 38.10%, and the epidemic durations are prolonged by 12.50 and 41.67%, respectively.

In summary, improving the intensity of one policy can effectively slow down the spread of the epidemic when the other two policies are fixed in a loose state. However, the effectiveness of the three policies is different: the RGOUT policy is the most effective in reducing peak value, delaying the peak arrival time and extending the epidemic duration; the second most effective is the RCONT policy, and the relatively weakest prevention and control policy is the RINFE policy.

In the third set of experiments (experiment 3-1 to experiment 3-6), policy effectiveness is compared by fixing the intensities of two types of policies in a strict state and varying the intensity of the other policy. When the intensities of the RCONT and RINFE policies are strict and the intensity of the RGOUT policy increases from loose to moderate, the infection curve is flattened (Figure 5): the peak value decreases by 58.22%, the peak arrival time is delayed by 81.25% and the epidemic duration is prolonged by 91.23%. Similarly, when the intensities of the RGOUT and RINFE policies are strict and the intensity of the RCONT policy increases from loose to moderate, the infection curve is flattened (Appendix A): the peak value decreases by 33.28%, the peak arrival time is delayed by 31.67%, and the epidemic duration is prolonged by 35.77%. When the intensities of the RGOUT and RCONT policies are strict and the intensity of the RINFE policy increases from loose to moderate, the infection curve is flattened (Appendix A): the peak value decreases by 34.95%, the peak arrival time is delayed by 32.05%, and the epidemic duration is prolonged by 7.46%.

In summary, improving the intensity of one policy can effectively slow down the spread of the epidemic when the other two policies are fixed in a strict state. However, the effectiveness of the three policies is different: the RGOUT policy is the most effective in reducing peak value, delaying the peak arrival time and extending the epidemic duration; the second most effective policies are the RCONT and RINFE policies.

### 3.3. Results of Basic Experiments Based on Different Kinds of Going-Out Needs

From the perspective of the speed of spread of the epidemic based on different kinds of going-out needs, the effect of controlling the physiological safety needs is better than the effects of controlling family and friendship needs. When the three types of policies are in a strict state, the flatness indices of the infection curve based on physiological safety needs are 64.37 and 55.23% of family needs and friendship needs.

In the fourth set of experiments (experiment 4-1 to experiment 4-3), this paper explores the impact of different going-out needs on the effectiveness of policy prevention and control. The infection curves of different kinds of going-out needs when the intensity of three policies is strict are presented in Figure 6, and the differences between infection curves are significant. Compared with physiological safety needs, the infection curves based on family needs and friendship needs can be flattened: the peak values decrease by 30.70 and 29.70%, while the peak arrival times are delayed by 27.34 and 11.51%, and the epidemic durations are extended by 7.65 and 27.30%. Similar results are found when the intensity of three policies is loose (Appendix A) and moderate (Appendix A).

The structure of family networks based on family needs and friend networks based on friendship needs are relatively stable, while stranger networks based on physiological safety needs are more random, which leads to more random contact and a more rapid spread of epidemics.

### 3.4. Results of the Comparison and Evaluation of Actual Epidemic Prevention and Control Policies

Through the improved curve similarity indicator based on dynamic time warping (DTW), we compared the real new daily infection data from 26 countries, regions, and cities around the world with the new daily infection data obtained by the basic experiments based on three kinds of going-out needs.

The similarity between the real and simulated data was calculated to obtain the curve similarity indicator. The smaller the calculated similarity of curves, the closer the policy implemented in the country, region, or city is to the corresponding experimental scenario. The results show that the countries, regions, and cities that are relatively close to Experiment 1-1 include Wuhan (China), the Republic of Korea, Germany, Norway, Russia, California (USA), and Australia; the countries, regions, and cities that are closer to Experiment 1-2 include Beijing (China), Japan, Singapore, the United Kingdom, Italy, New York (USA), Canada, and Egypt; and the countries, regions, and cities that are closer to Experiment 1-3 include Czech, Belgium and Peru.

Appendix A show the new daily infections of 19 countries, regions, and cities. Among them, Wuhan, China, the Republic of Korea, Norway, Germany, Spain, and Australia may have adopted stricter epidemic prevention and control policies when the number of new daily infections starts to increase, allowing the curve to reach a peak in a short period of time and to bend more quickly. Countries such as Japan, Singapore, Pakistan, Hungary, Canada, Egypt, and South Africa may have adopted relatively loose policies when the number of new daily infections begins to increase. The process of reaching the peak takes a long time, but after reaching the peak, it can also effectively reduce the number of infections.

### 3.5. Robustness Test

To explore the influence of individual quantity on the results, this paper also provides the infection curves of basic experiments based on three kinds of going-out needs with 4000 and 5000 individuals (Figure 7 and Figure 8). There is an obvious difference among the three infection curves, and the results are consistent with the results when the number of individuals is 2000. It is proven that the conclusion of this paper has a certain stability and can be applied to larger networks.

## 4. Discussion

From the perspective of individuals going out, we built a social network based on the going-out needs of individuals and combined it with the SEIR model to simulate the process of individuals going out and the spread of epidemics. At the same time, various prevention and control policies were classified and parameterized to act on different links in the individual going out process to realize the comparison and evaluation of policy effectiveness. In the basic experiments based on the three kinds of going-out needs, the epidemic ends in a network consisting of 2000 individuals in 428 days when the intensity of all three types of policy is strict, and the infection curve is quite flat. However, when the intensity of all three policies is moderate and loose, the epidemic ends at 119 and 48 days, respectively, and the infection curve becomes steeper. Therefore, if the intensity of prevention and control policies is as strict as possible, the infection curve can be flattened and the spread of the epidemic can be slowed. This is consistent with the conclusions of some existing studies [33,34,39]. At the same time, the number of new daily infections can be kept at a low level, ensuring that limited medical resources can be put to good use. 

In the variation experiments based on the three kinds of going-out needs, when keeping the intensity of two types of policies fixed and only varying the intensity of the other policy, it can be found that the stricter the varied policy, the more effective it is in preventing and controlling the epidemic, with the RGOUT policy being the most effective. When it is difficult to control the intensity of all policies to a very strict level, the most effective RGOUT policy can be prioritized.

In the basic experiments based on different kinds of going-out needs, under the same policy intensity conditions, going out based on physiological safety needs increases the randomness of contacts in the network, which in turn accelerates the spread of the epidemic, especially when all three policy intensities are all in a strict state. Going out based on physiological safety needs is not conducive to epidemic prevention and control, but this is a basic need that has to be met. Therefore, some tasks, such as purchasing daily necessities, can be carried out by community volunteers, which can effectively reduce random contact.

This paper still has some shortcomings. First, in terms of going-out needs, only family needs, friendship needs and physiological safety needs are considered for the time being, while the needs for school and work have yet to be taken into consideration. Second, the number of individuals simulated in this paper is only 2000. Since individual-based models are often time intensive and computationally expensive to implement, requiring a high degree of expertise and computational resources [51], simulation experiments have not been conducted with a larger number of individuals, except for the robustness test. Third, this paper does not consider asymptomatic infections and some policy such as contact tracing.

## 5. Conclusions

In summary, this paper innovatively parameterizes nonpharmacological prevention and control policies combined with social network and SEIR models from the perspective of individuals going out behavior based on human needs to simulate the process of individuals going out and the spread of epidemics. We realize the comparison and evaluation of the effectiveness of different intensities of the same type of policy and of different types of policies. The simulation results show that: (1) enhancing policy implementation can effectively prevent and control the epidemic; (2) among the different types of policies, policies that limit the time of going out are most effective; and (3) the effect of controlling the physiological safety needs is better than the effects of controlling family and friendship needs. 

The experimental results of this paper will have important theoretical and practical implications for the formulation of epidemic prevention and control policies, the selection of intensity and combination decision-making. Future studies could include more individuals and consider asymptomatic infections and some policy such as contact tracing when studying the effectiveness of policies.

## Figures and Tables

**Figure 1 ijerph-19-16222-f001:**
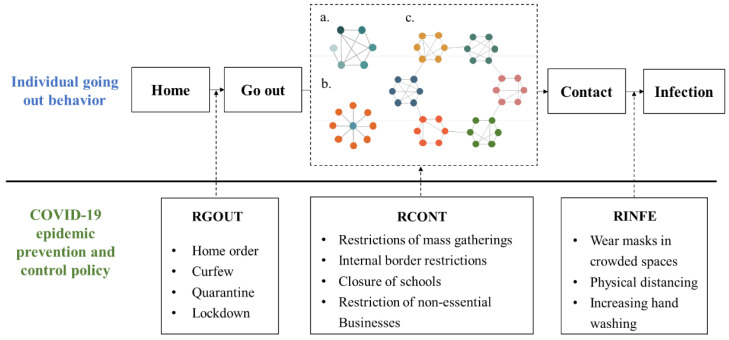
Mechanism diagram based on individual behavior. Note: (a) represents a family network based on family needs; (b) represents a stranger network based on physiological safety needs; (c) represents a friend network based on friendship needs.

**Figure 2 ijerph-19-16222-f002:**
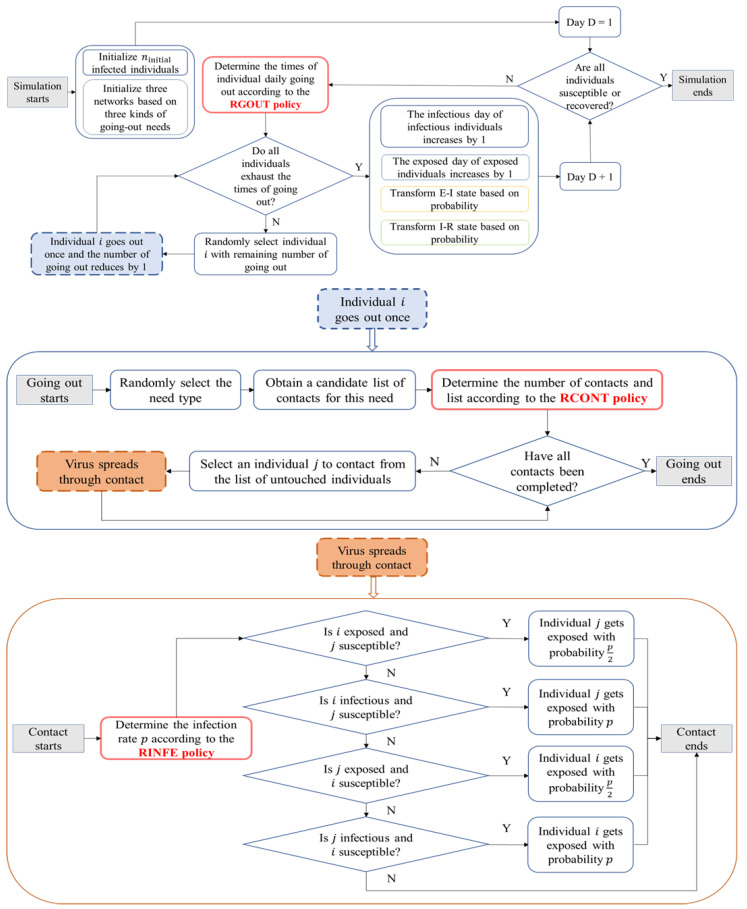
Simulation process.

**Figure 3 ijerph-19-16222-f003:**
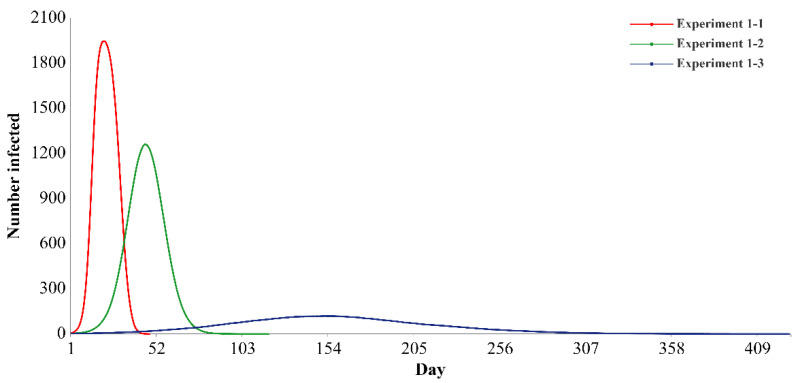
Infection curves of basic experiments based on three kinds of going-out needs. Note: Experiments 1-1 to 1-3 indicate that the three types of policies are all in strict, moderate and loose states, respectively. Specific parameter values are presented in Table 2.

**Figure 4 ijerph-19-16222-f004:**
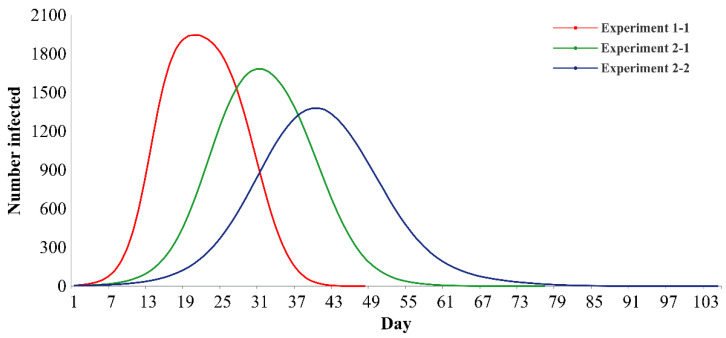
Infection curves of different intensities of the RGOUT policy (the other policies are loose).

**Figure 5 ijerph-19-16222-f005:**
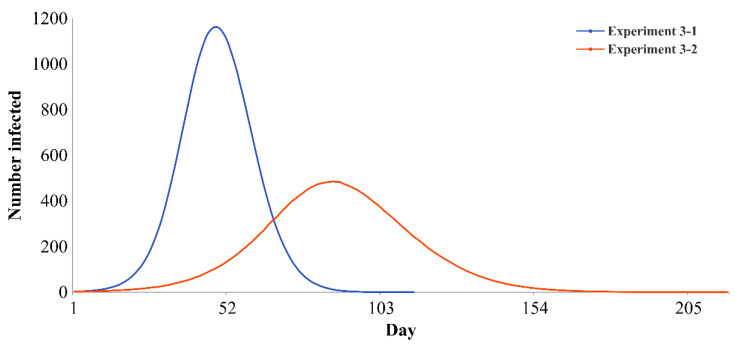
Infection curves of different intensities of the RGOUT policy (the other policies are strict).

**Figure 6 ijerph-19-16222-f006:**
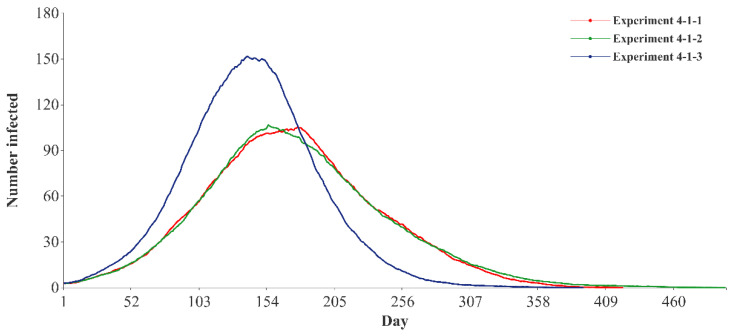
Infection curves of different kinds of going-out needs with strict policy intensity.

**Figure 7 ijerph-19-16222-f007:**
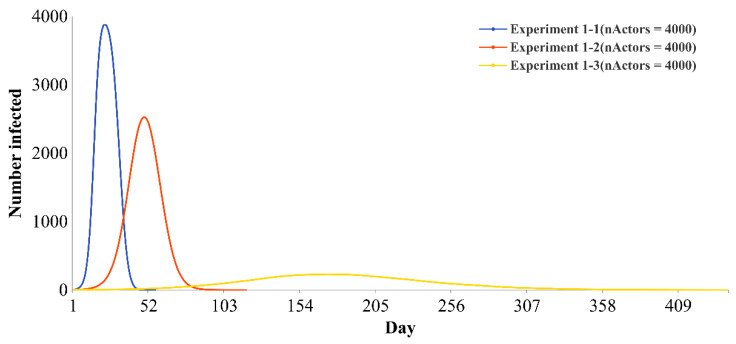
Infection curves of the basic experiments with 4000 individuals.

**Figure 8 ijerph-19-16222-f008:**
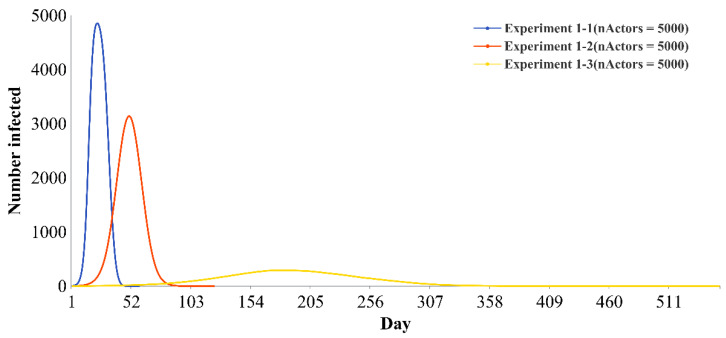
Infection curves of the basic experiments with 5000 individuals.

**Table 1 ijerph-19-16222-t001:** Quantitative design strategies for prevention and control policies.

Policies	Loose	Moderate	Strict
RGOUT policy (Daily Go Out Times Threshold)	2	1	0.5
RCONT policy (Daily Every Time Contact Threshold)	6	4	2
RINFE policy (Infection Rate)	0.15	0.12	0.09

**Table 2 ijerph-19-16222-t002:** Basic experiments based on three kinds of going out needs.

Experimental Scenarios	Daily Go Out Times Threshold	Daily Everytime Contact Threshold	Infection Rate
Experiment 1-1	2	6	0.15
Experiment 1-2	1	4	0.12
Experiment 1-3	0.5	2	0.09

**Table 3 ijerph-19-16222-t003:** Variation experiments based on three kinds of going-out needs.

Experimental Scenarios	Daily Go Out Times Threshold	Daily Everytime Contact Threshold	Infection Rate
Experiment 2-1	1	6	0.15
Experiment 2-2	0.5	6	0.15
Experiment 2-3	2	4	0.15
Experiment 2-4	2	2	0.15
Experiment 2-5	2	6	0.12
Experiment 2-6	2	6	0.09
Experiment 3-1	2	2	0.09
Experiment 3-2	1	2	0.09
Experiment 3-3	0.5	6	0.09
Experiment 3-4	0.5	4	0.09
Experiment 3-5	0.5	2	0.15
Experiment 3-6	0.5	2	0.12

**Table 4 ijerph-19-16222-t004:** Basic experiments based on different kinds of going-out needs.

Experimental Scenarios	Policy Intensity	Sub-Experiment Scenarios	Going-Out Needs
Experiment 4-1	The intensity of all three policies is strict	Experiment 4-1-1	Family needs
		Experiment 4-1-2	Friendship needs
		Experiment 4-1-3	Physical safety needs
Experiment 4-2	The intensity of all three policies is loose	Experiment 4-2-1	Family needs
		Experiment 4-2-2	Friendship needs
		Experiment 4-2-3	Physical safety needs
Experiment 4-3	The intensity of all three policies is moderate	Experiment 4-3-1	Family needs
		Experiment 4-3-2	Friendship needs
		Experiment 4-3-3	Physical safety needs

## Data Availability

Not applicable.

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
