# Peer review of "Suit the Remedy to the Case—The Effectiveness of COVID-19 Nonpharmaceutical Prevention and Control Policies Based on Individual Going-Out Behavior"

_ijerph, 2022, doi:10.3390/ijerph192316222_

Round 1

Reviewer 1 Report

1. These assumptions should be supported by the scientific evidence (literature etc) - lines 76 -78 - "We assume that individuals have three kinds of needs: family needs (meeting with family),  social needs (connecting with friends), and physiological safety needs (purchasing necessities, etc.)".

2. Information on the similar research is missed

3. List of references is limited to 39 positions

4. The paper is well structured and the research is interesting.

Reviewer 2 Report

I have strong doubts about the manuscript. I think longer time is need to judge the nonpharmaceutical prevention and control policies against COVID.

I am willing to read again only if the paper will be improved substantially with scientific facts.

Reviewer 3 Report

The manuscript evaluates COVID-19 prevention and control policies, specifically the individual going out behavior was assessed through family needs, social needs and physiological safety needs. A model of individual going out and epidemic spread were simulated and tested on data collected during pandemics which are the effectiveness in reducing infection according to the policy applied.  The strict control of polic ywas found to have highest inhibitory effect on decreasing the peak value, delaying the peak arrival time. These kind of simulation studies were suggested to be useful in policy decision. I think the experimental approach and the results supports the conclusions and discussions.

Reviewer 4 Report

The authors propose a model to measure the effectiveness of different types of policies for the prevention and control of COVID-19. In this study, three kinds of networks have been considered based on family needs, social needs, and physiological safety needs. Thereafter, a model of individuals going out and spreading epidemics using a simulation method is used to simulate various scenarios. The proposed model determines the effectiveness of the policy in preventing COVID-19 outbreaks. But there are some suggestions that need to be added to the manuscript before they can be taken into account.

1. The abstract needs to be more interesting and informative about the used statistical method.

2. The manuscript needs to include more works of literature that are related to the proposed model and their research gaps. 

3. How many policies have been considered in this study?

4. The statistical outcome from the proposed model needs to be compared with the existing model.  

5. What are the limitation of the proposed model ?

6. Highlights of the proposed work need to incorporated

7. What is novelty of the proposed model?

Round 2

Reviewer 1 Report

It has been improved after some Reviewers comments. 

Author Response

Thank you very much!

Reviewer 4 Report

Authors have included all the suggestions suggested by the reviewer

Author Response

Thank you very much!